# Single and Joined Behaviour of Circulating Biomarkers and Metabolic Parameters in High-Fit and Low-Fit Healthy Females

**DOI:** 10.3390/ijms24044202

**Published:** 2023-02-20

**Authors:** Joëlle J. E. Janssen, Bart Lagerwaard, Arie G. Nieuwenhuizen, Xavier Escoté, Núria Canela, Josep M. del Bas, Vincent C. J. de Boer, Jaap Keijer

**Affiliations:** 1Human and Animal Physiology, Wageningen University, P.O. Box 338, 6700 AH Wageningen, The Netherlands; 2Cell Biology and Immunology, Wageningen University, P.O. Box 338, 6700 AH Wageningen, The Netherlands; 3TI Food and Nutrition, P.O. Box 557, 6700 AN Wageningen, The Netherlands; 4EURECAT, Technology Centre of Catalunya, Nutrition and Health Unit, 43204 Reus, Spain; 5Eurecat Centre Tecnològic de Catalunya-Centre for Omic Sciences (COS), Joint Unit University of Rovira i Virgili-EURECAT, 43204 Reus, Spain

**Keywords:** circulating biomarkers, health, lifestyle, aerobic fitness level, exercise, human, females

## Abstract

Biomarkers are important in the assessment of health and disease, but are poorly studied in still healthy individuals with a (potential) different risk for metabolic disease. This study investigated, first, how single biomarkers and metabolic parameters, functional biomarker and metabolic parameter categories, and total biomarker and metabolic parameter profiles behave in young healthy female adults of different aerobic fitness and, second, how these biomarkers and metabolic parameters are affected by recent exercise in these healthy individuals. A total of 102 biomarkers and metabolic parameters were analysed in serum or plasma samples from 30 young, healthy, female adults divided into a high-fit (V̇O2peak ≥ 47 mL/kg/min, N = 15) and a low-fit (V̇O2peak ≤ 37 mL/kg/min, N = 15) group, at baseline and overnight after a single bout of exercise (60 min, 70% V̇O2peak). Our results show that total biomarker and metabolic parameter profiles were similar between high-fit and low-fit females. Recent exercise significantly affected several single biomarkers and metabolic parameters, mostly related to inflammation and lipid metabolism. Furthermore, functional biomarker and metabolic parameter categories corresponded to biomarker and metabolic parameter clusters generated via hierarchical clustering models. In conclusion, this study provides insight into the single and joined behavior of circulating biomarkers and metabolic parameters in healthy females, and identified functional biomarker and metabolic parameter categories that may be used for the characterisation of human health physiology.

## 1. Introduction

Lifestyle factors play a dominant role in health maintenance and the prevention of chronic diseases, such as type 2 diabetes [1], cardiovascular disease [2], and cancer [3]. Adopting a healthy lifestyle, including regular physical activity is associated with a lower chronic disease risk [4]. Biomarkers and metabolic parameters, which together are considered measurable indicators of biological states or conditions, are important to evaluate the impact of interventions on health status [5]. To monitor health status, biomarkers that reflect overarching physiological processes, such as metabolism, inflammation, and oxidative stress, have been proposed [6]. Their application in monitoring the health-to-disease trajectory depends on their ability to detect disease, but also on their behaviour in pre-disease conditions. For example, how these biomarkers behave in healthy individuals adhering to different lifestyles and whether they respond to short-term perturbations. Moreover, many of these biomarkers and metabolic parameters have not been studied relative to each other, especially not in healthy individuals. 

Physical activity is one of the lifestyle factors that has been linked to a reduced chronic disease risk [7]. High levels of physical activity, reduced insulin resistance [8], improved lipoprotein profiles [9], and lowered interleukin (IL)-6 levels in the long-term [10], contribute to a lower chronic disease risk [11,12,13,14]. It is not well studied whether systemic changes are already visible in the basal state in healthy individuals with different habitual physical activity levels, reflected in differences in aerobic fitness. It is known that short-term exercise, e.g., a single bout of exercise, provokes acute systemic changes, which can last for up to 24 h [15,16,17]. It may be possible that these short-term exercise responses differ between individuals with high and low levels of aerobic fitness, due to the metabolic and physiological adaptations of the body to regular exercise [18,19]. This, however, has been studied only to a very limited extent, with most studies focusing on male individuals [15], while physiological responses between males and females can be strikingly different [20]. 

Here, we investigated whether serum and plasma biomarkers associated with overarching physiological processes are affected in the basal state and after a recent bout of exercise in healthy, young female adults differing in aerobic fitness level, implicating a potentially different metabolic disease risk. The original objectives of the study were to investigate whether differences exist in vitamin B2 status, in muscle mitochondrial capacity, and in circulating cellular and molecular parameters between high aerobically fit (high-fit) and low aerobically fit (low-fit) females with a validated difference in V̇O_2_peak, and after a recent bout of exercise. Previously, we found a significant difference in skeletal muscle mitochondrial oxygen consumption rate [21] and mitochondrial function in white blood cells [22] between the high-fit and low-fit females in our healthy study population. Circulating biomarker and metabolic parameter analysis in this study will now show whether single biomarkers, functional biomarkers, metabolic parameter categories, total biomarkers, and metabolic parameter profiles differ between high-fit and low-fit females, i.e., whether the biomarkers and metabolic parameter, including those that discriminate between health and disease, already differ in healthy individuals with potentially a different long term disease risk, in the basal state and after a recent bout of exercise. This information is a prerequisite for the faithful application of these biomarkers and metabolic parameters in preventative and health improvement interventions. 

## 2. Results

### 2.1. Biomarker and Metabolic Parameter Analysis Was Reproducible across Analysis Platforms 

All 102 biomarkers were analysed in samples from a well-characterised study [21,22,23] of healthy females. This study population represents high-fit (V̇O_2_peak ≥ 47 mL/kg/min, N = 15) and low-fit (V̇O_2_peak ≤ 37 mL/kg/min, N = 15) females (Table 1), which was supported by a significantly higher skeletal muscle mitochondrial capacity for ATP generation [21] and a better mitochondrial function, including higher basal and uncoupled oxygen consumption rate, in peripheral blood mononuclear cells (PBMCs) [22] in the high-fit compared to the low-fit females. 

Both groups were assessed at baseline and 22 h after a single bout of exercise. To establish the reproducibility of the biomarker and metabolic parameter determination, a random subset of 16 biomarkers and metabolic parameters involved in protein and lipid metabolism, was also determined using a similar ^1^H NMR technology, but with different matrices and laboratories. Significant correlations were observed for all 16 markers, with correlation coefficients between 0.59–0.90 for the amino acids (all eight *p* < 0.0001), 0.42–0.65 for the six fatty acids (one *p* < 0.01, three *p* < 0.001, two *p* < 0.0001), and 0.63 and 0.86 for the two ketone bodies (both *p* < 0.0001, Appendix A). This supports the validity of these biomarker measurements and demonstrates the robustness of our approach.

### 2.2. Single Biomarker and Metabolic Parameter Analysis Demonstrates a Similar Biomarker Profile between High-Fit and Low-Fit Females

To better evaluate which functional processes are affected by alterations in biomarker and metabolic parameter levels, we first linked each biomarker and metabolic parameter to one of the following physiological processes: hormone signaling, inflammation and oxidative stress responses, and metabolism. This resulted in three overarching, functional biomarkers and metabolic parameter categories: (1) peptide hormones (Appendix A), (2) inflammation and oxidative stress responses (Appendix A), and (3) metabolism, which was further divided into protein, carbohydrate, and lipid metabolites (Appendix A). Since many biomarkers were related to lipid metabolism, this subcategory was further subdivided into fatty acids, cholines, ketone bodies, acylcarnitines, cholesterol metabolites, and lipoproteins. All mean or median biomarker values and ranges for high-fit and low-fit females at baseline and after recent exercise are in Appendix A, which provides all the results in a comprehensive manner. To assess the effect of fitness level and the recent bout of exercise on the individual biomarker and metabolic parameter responses, RM-ANOVA on the raw data (for normally distributed biomarkers and metabolic parameters) or transformed data (for not normally distributed biomarkers and metabolic parameters) was performed. This resulted in a fitness level effect (rawP_group_), a recent exercise effect (rawP_exercise_), and an interaction effect (rawP_group*exercise_) for each biomarker and metabolic parameter, and these raw *p*-values were corrected for multiple testing, with a significance cut-off of <0.10 for adj.P_group_, adj.P_exercis_, and adj.P_group*exercise_. The detailed results of these analyses and the measure of effect size (partial eta squares) are in Appendix A and an overview is presented in Figure 1. None of the individual biomarkers and metabolic parameters were significantly impacted by fitness level, except for the ‘peptide hormone’ leptin (Figure 1), which was significantly higher in low-fit females compared to high-fit females (adj.P_group_ = 0.076, Figure 2A), in line with their different adiposity, supporting data validity. None of the markers and metabolic parameters related to inflammation, oxidative stress, or metabolism was significantly impacted by fitness level in our healthy females, indicating that healthy high-fit and low-fit females have similar biomarker and metabolic parameter profiles. 

### 2.3. Recent Exercise Regulates Single Biomarkers and Metabolic Parameter Related to Inflammation, Lipid Metabolism, and Hormone Signaling 

Next, we assessed whether recent exercise altered individual biomarkers and metabolic levels, and examined whether high-fit and low-fit females responded differently to recent exercise. Recent exercise significantly regulated 35 of the 102 biomarkers and metabolic parameters. These biomarkers and metabolic parameters were related to hormone signaling, inflammation and oxidative stress, lipid metabolism, and protein metabolism (Figure 1). The peptide hormone adiponectin was significantly increased after exercise in both groups (adj.P_exercise_ = 0.001, Figure 2B). Of the 10 biomarkers and metabolic parameters that are related to inflammation and oxidative stress, seven were significantly regulated by exercise, the top five being N-acetylglycoproteins (up; adj.P_exercise_ = 4.16 × 10^−6^), MCP1 (down, adj.P_exercise_ = 4.16 × 10^−6^), TNF (down, adj.P_exercise_ = 3.09 × 10^−4^), CRP (up, adj.P_exercise_ = 0.003), and IL10 (down, adj.P_exercise_ = 0.003, Figure 2C–G). In total 27 metabolic markers and metabolic parameters were significantly regulated by exercise, with the top five all linked to lipid metabolism, with increased levels of lysophosphatidylcholine (adj.P_exercise_ = 3.51 × 10^−6^, Figure 2H), and increased levels of apolipoprotein A1, total esterified cholesterol, total cholesterol, and HDL cholesterol (Figure 2I–L, all adj.P_exercise_ = 0.001). Importantly, for none of the 102 biomarkers, the exercise response differed significantly between high-fit and low-fit females (all adj.P_group*exercise_ > 0.10). We, therefore, performed an additional main effect analysis without the interaction term, which resulted in the same significantly regulated biomarkers and metabolic parameters as compared to the full interaction model, except for MUFA (adj.P_exercise_ = 0.101, Appendix A). 

In summary, this single biomarker and metabolic parameters analysis demonstrated that various biomarkers and metabolic parameters linked to inflammation, lipids, protein metabolism, and adiponectin were significantly regulated by recent exercise, while only leptin was affected by fitness level in these healthy females (Figure 3).

### 2.4. Data-Driven Biomarker and Metabolic Parameter Clusters Link with Functional Biomarker and Metabolic Parameter Categories

Next, we studied the joined dynamics of these biomarkers and metabolic parameters. Hierarchical clustering was applied on the scaled biomarker and metabolic parameters levels in all four groups and visualised in a heatmap (Figure 4). The heatmap generated multiple biomarker and metabolic parameters clusters that corresponded to our predefined functional biomarker and metabolic parameters categories, indicated by clustering of inflammation and oxidative stress-related markers, amino acids, fatty acids, ketone bodies, acylcarnitines, lipoproteins, and cholesterol metabolites along the *y*-axis (Figure 4). Although some of these functional biomarkers and metabolic parameters categories also displayed *x*-axis clustering (e.g., the lipoproteins and fatty acids), the overall heatmap pattern was only slightly related to fitness level and not related to recent exercise. Instead, the intra-individual biomarker and metabolic parameters response, i.e., baseline and post-exercise values within one subject, accounted for most of the *x*-axis clustering. The notion that biomarker and metabolic parameters levels were primarily affected by interindividual differences, rather than by fitness level or the recent bout of exercise, was confirmed by a principle component analysis (PCA), where no clear separation was observed between our experimental conditions (Appendix A). To obtain a more detailed understanding of data-driven relationships between biomarkers and metabolic parameters, a hierarchically clustered (*p* < 0.05) correlation matrix was generated (Figure 5), with significant Spearman r > 0.6 or <0.6 correlations indicated as potential physiological relevant links (Figure 5). As above, these data-driven correlations corresponded to functional categories, such as amino acids (especially the branched-chain amino acids (BCAAs)), fatty acids, ketone bodies, acylcarnitines, cholesterol metabolites, and lipoproteins (Figure 5). However, some data-driven correlated biomarkers and metabolic parameters were not in line with our predefined functional biomarker and metabolic parameters categories, such as CRP and glycine (r = −0.72), glutamine and hydroxyisovalerylcarnitine (C5:0-OH, r = 0.60), tyrosine and hydroxyisovalerylcarnitine (C5:0-OH, r = 0.64), tyrosine and methylcrotonylcarnitine (C5:1, r = 0.66), betaine and octadecadienylcarnitine (C18:2, r = 0.65), and N-acetylglycoproteins and lysophosphatidylcholine (r = 0.65), all having a *p* < 1.0× 10^−7^. Several markers showed particular strong, independent correlations (*p* < 1.0× 10^−9^), such as TNFA and MCP1, CRP and glycine, isoleucine, leucine and valine, 3-hydroxybutyrate, acetoacetate and acetone; a subgroup of fatty acids (SCFA, total FA, Omega3 FA, Linoleic acid, MUFA, Oleic acid, PUFA, and ARA&EPA); two subgroups of carnitines (≤C5; ≥C6) and cholesterols containing lipids (Appendix A). Of note, similar patterns were observed when not all the groups, but only the baseline levels of high-fit and low-fit females, were included (Appendix A). Overall, this integrated biomarker and metabolic parameters analysis demonstrated that data-driven biomarkers and metabolic parameter clusters are composed of biomarkers and metabolic parameters that are functionally linked and that these clusters largely correspond with our predefined functional biomarkers and metabolic parameters categories. 

## 3. Discussion

We performed an elaborate analysis of 102 circulating biomarkers and metabolic parameters, previously studied in disease conditions such as diabetes type 2, obesity, and cardiovascular disease [24,25,26,27], but hardly in healthy individuals with different lifestyles. Analysis of a selection of these biomarkers and metabolic parameters across two platforms showed similar results, underpinning their reliability, and indicating the robustness of these platforms. Except for leptin, individual biomarkers, and metabolic parameters, levels were not significantly different between high and low aerobically fit females. Since leptin levels have been positively correlated to body fat percentage [28,29], the difference in leptin presumably results from a significant difference in body fat percentage between high-fit and low-fit females, further underpinning the validity of our data. Our observation that all other biomarkers and metabolic parameters were similar between the two groups, while previous studies in high and low aerobically fit individuals found significant differences in e.g., lipid and protein metabolites [30,31,32,33,34,35], is likely related to our standardised experimental set-up, as compared to other studies. We studied healthy, young-adult females of similar age and BMI in a highly controlled setting, while previous studies were performed with metabolically impaired individuals [26], and individuals with substantially different BMIs [30,31,32,35], or wider age ranges [30,35] in experimental conditions that were less standardised [30,31,32,33,35], and these factors especially impact circulating metabolite levels [26,30,31]. Given that the levels of the analysed biomarkers and metabolic parameters were similar among the healthy females in our study, and multiple of these biomarkers and metabolic parameters showed dysregulation during disease, our findings imply that this biomarker and metabolic parameters set could be used to monitor progress from a healthy to an unhealthier state and may be used in health improvement interventions. Previously, we reported that maximal and basal respiration of the peripheral blood mononuclear cell (PBMC) fraction was different between the high-fit and low-fit individuals that were studied here, but there was no additional effect of exercise [22]. This suggests that PBMC oxidative metabolism may be a more sensitive fitness biomarker and metabolic parameter than the circulating metabolites studied here.

Studies that focus on recent exercise effects, i.e., effects on the day after exercise completion, are scarce compared to studies on acute or chronic exercise [15,18]. Yet, recent exercise is especially relevant for biomarkers and metabolic parameters, as they can indicate whether the physical activity of the study’s subjects should be controlled prior to sampling. Here, we demonstrated that adiponectin, lipid metabolites, and inflammatory markers were most responsive to recent exercise, which is in line with data from other studies [36,37,38,39]. These findings suggest that future biomarker and metabolic parameter studies should consider the standardisation of study subjects’ physical activity at least 24 h prior to blood sampling, especially when they include hormones and markers related to lipid metabolism and inflammation. 

Multiple separate clusters that were obtained in the heatmap and correlation matrix included biomarkers and metabolic parameters that corresponded to biomarkers and metabolic parameters embedded in our predefined, functional biomarker and metabolic parameter categories. Examples are the BCAAs, fatty acids, ketone bodies, short-chain acylcarnitines, long-chain acylcarnitines, cholesterol metabolites, and lipoproteins, which suggests that the response of biomarkers and metabolic parameters within these (sub)categories are interdependent. This has two important implications. First, one biomarker and metabolic parameters within a cluster could be considered representative of the total cluster (e.g., isoleucine for the BCAAs), which could be of relevance for studies that measure only one or a limited number of biomarkers and/or metabolic parameters from one correlated cluster. Second, it provides opportunities for future studies to compute one total, standardised score for all is within a cluster that are strongly correlated (e.g., a total BCAA score). From a disease risk assessment point of view, such an integrated score will likely have a larger power and stronger predictive value as compared to individual biomarker levels. Previously, Wang et al. have found that BCAA levels could predict diabetes type 2 risk [24]. Integrated BCAA analysis is therefore promising as health-status biomarker. Not all biomarkers from functional categories can be integrated because of differences in the individual responses (e.g., peptide hormones, inflammation markers, and short- versus longer-chain acylcarnitines). Clustering outside the functional category was also observed. The inverse association between CRP and the amino acid glycine has also been demonstrated previously [40,41] and likely results from the inflammation-modulating capacity of glycine [42,43]. The positive association between N-acetylglycoprotein and lysophosphatidylcholine is also likely mediated via inflammation since N-acetyglycoproteins plasma levels correlate with lipoprotein-associated phospholipase A2 levels [44], which generates lysophosphatidylcholine to promote inflammation [45,46]. Direct positive links between glutamine, tyrosine, C5:0-OH, and C5:1 acylcarnitines have not yet been described but could be mediated by BCAA breakdown [47,48]. The positive link between betaine and C18:2 acylcarnitine has not yet been demonstrated in humans but may be related to fatty acid incorporation, as previously demonstrated in pigs [49]. The observed correlations imply some revision of our a-priory functional categorisation and, importantly, provide leads for biomarker and metabolic parameter integration and functional interpretation of changes in biomarker and metabolic parameter levels, especially the observed sets of highly correlated markers arising from this non-supervised correlation analysis (Appendix A). 

Next to the functional links between biomarker pairs, the hierarchical clustering models also showed that the degree of clustering for the intraindividual biomarker and/or metabolic parameters response i.e., the baseline and post-exercise biomarker or metabolic parameters values of one subject, was higher than the degree of clustering of the group (high-fit versus low-fit) and the timepoint (baseline versus post-exercise) biomarker and metabolic parameters responses. This finding suggests a considerable level of interindividual variation in our study population, which might also explain our observation that ~35% of the biomarkers and metabolic parameters were significantly impacted by recent exercise, but that clustering did not separate total baseline and post-exercise biomarker and metabolic parameters profiles. Since Krug et al., also showed that the interindividual variability was increased by using challenge tests [50], one could speculate that the challenged biomarker and metabolic parameters responses within one individual over time might act as a better predictor of health status, as compared to a singular analysis of the average biomarker and metabolic parameters levels of a larger group during basal homeostasis. 

Our study included some strengths and limitations. One of the strengths is the integrated approach to analyse single as well as joined biomarker and metabolic parameters behavior in a healthy, homogenous study population at basal as well as challenged conditions, which provided us with better insight into the behaviour of biomarker and metabolic parameters, relative to each other. An understanding of biomarkers in healthy individuals is a prerequisite for their use in preventive health, for example, biomarkers and metabolic parameters guided dietary advice for health improvement. Another strength of our study is the focus on female individuals, since sex can affect metabolic responses [20,30], and females are often underrepresented in biomarker studies [15]. One limitation of our study is the relatively small sample size with mixed Caucasian and non-Caucasian healthy females. Nevertheless, it allowed us to identify PBMC mitochondrial oxygen consumption rate as a biomarker and metabolic parameter for differences in fitness in healthy females [22], while here we established the sensitivity of circulating biomarkers and metabolic parameters to a recent bout of exercise. Another limitation of our study is that we could not determine the contribution of intraindividual variation, i.e., the day-to-day variation within an individual, as we sampled only twice in a relatively short time span. Although previous studies have demonstrated that the intraindividual variation for circulating adipokines [51], inflammatory markers [51,52], and metabolites [53,54] is smaller than the interindividual variation, we cannot exclude this source of error in our study. Third, we did not include additional post-exercise sampling time points, e.g., immediately post-exercise or a few hours post-exercise, nor established the post-exercise dynamics of the biomarkers and metabolic parameters. The levels of most inflammatory markers, oxidative stress-related markers, and metabolites change acutely or in the first few hours after exercise, with each marker having its own kinetic profile [15,18], biomarker, and metabolic parameter kinetics can also differ between individuals as a result of interindividual variation [50], therefore sampling at multiple timepoints after the exercise bout would have given insight in the exercise-induced biomarker and metabolic parameters behavior with time. Here, we provided insight into which biomarkers and metabolic parameters were still not back to baseline after 22 h after exercise. Fourth, our study focused on a total of 102 biomarkers and metabolic parameters related to hormone signaling, inflammation and oxidative stress, and metabolism, while fitness level and single exercise stimulation have been associated with alterations in markers that were not included in our study, such as vitamins [32,33], ceramides [26], individual lysophosphatidylcholines [26,30] and bile acids, which could possibly have provided additional insights in these biomarkers and metabolic parameters in view of the homogeneity of our study subjects characteristics and high level of study standardisation.

In conclusion, we showed that the overall circulating biomarker and metabolic parameters profiles were similar in young adult females with different aerobic fitness levels, i.e., with potentially different disease risks, in the healthy state. Recent exercise significantly affected a selected number of individual biomarkers and metabolic parameters but was not dependent on fitness level. This study provides insight into the single and joined behaviour of circulating biomarkers and metabolic parameters in healthy females, and identified functional biomarker and metabolic parameter categories that may be used for the characterisation of human health physiology.

## 4. Materials and Methods

### 4.1. Ethical Approval and Study Registration

The protocol for collection and handling of human samples was ethically approved by the medical ethical committee (METC) of Wageningen University (since January 2021 replaced by METC Oost-Nederland) with reference number NL70136.081.19 and registered in the Dutch trial register (NL7891) on 23 July 2019. All procedures performed were in accordance with institutional ethical standards, national law (WMO, The Hague, 1998), and with the 1964 Helsinki declaration and its amendments. Written informed consent was obtained from all individual subjects included in the study. 

### 4.2. Study Subjects

Healthy young females (18–28 years of age, BMI 18.5–25 kg/m^2^) were recruited from the local university and community population. Exclusion criteria were as follows: history of cardiovascular, respiratory, haematological, or metabolic disease; use of prescribed chronic medication; anaemia (haemoglobin concentration < 12 g/dL); blood donation within two months before the start of the study; smoking (>5 cigarettes per week); recreational drug use, or over the counter drug use during the study; use of performance-enhancing supplements; pregnancy or lactating. Subjects were selected if they had a V̇O_2_peak ≥ 47 mL/kg/min (high-fit group), or ≤ 37 mL/kg/min (low-fit group) determined using a maximal exercise test, measured using the validated screening protocol of Lagerwaard et al. [21,55], which minimised the risk for selective bias. The cut-off for VO_2_-peak can be found in [21]. Sixteen high-fit and sixteen low-fit subjects were included. The V̇O_2_peak data and results of skeletal muscle mitochondrial capacity of these subjects have been published previously by our group [21]. A total of 111 maximal exercise tests were performed to end up with the desired sample size. One subject was excluded due to medication intake and one subject was excluded due to symptoms of illness directly after completion of the study protocol. The use of oral contraceptives was not excluded; only the use of monophasic oral contraceptives containing low synthetic oestradiol and progesterone was allowed and was controlled for (N = 7 in the high-fit and N = 6 in the low-fit group). The 17ß-estradiol levels were measured using a chemiluminescent immunoassay on a Lumipulse G1200 analyser (Fujirebio Incl) at the Erasmus Medical Centre (Rotterdam, the Netherlands) and were not significantly different between those high-fit (527.7 [353.1–610.0]) and low-fit females (217.4 [109.1–895.2]) that did not use oral OC (*p* = 0.321). 

### 4.3. Study Design

Subjects participated within the end of their follicular phase until menstruation, based on self-reported menstruation or during the final 14 days of the contraceptive-cycle (contraceptive usage was equal between the groups). Subjects refrained from heavy physical exercise 48 h prior to the first study day and any physical exercise and alcohol consumption 24 h prior to the first study day. Subjects adhered to dietary guidelines 24 h prior to each study day, which included the consumption of a standardised evening meal (73% carbohydrates/16% protein/11% fat, 1818 kJ) before 8:00 PM and dietary guidelines for the consumption of breakfast, lunch, drinks, and snacks. After an overnight fast, blood was collected on the morning of the first study day (= baseline time point) and on the morning of the second study day, i.e., 21 h after a single bout of exercise (= post-exercise time point). Blood samples (3 × 6 mL) were collected by venipuncture in vacutainers containing dipotassium EDTA (K2-EDTA) as an anticoagulant for plasma collection (2 × 6 mL, BD Biosciences, Vianen, Netherlands) and a vacutainer containing silica as a clot activator for serum collection (1 × 6 mL, BD Biosciences, Vianen, The Netherlands). Blood tubes for plasma collection were kept on ice-water and processed within 30 min after blood collection. Blood tubes for serum collection were kept at RT for 60 min to allow clotting and immediately processed afterwards. Body fat percentage was measured according to the four-site method by Durnin-Womersley using the skinfold measurements of the triceps, biceps, sub scapula, and supra iliac, measured using a skinfold calliper (Harpenden, UK). Subjects received breakfast and after two hours, subjects completed an individualised exercise protocol consisting of 60 min cycling on an electrically braked bicycle ergometer (Corival CPET, Lode, The Netherlands) at a workload aiming to equal 70% of their V̇O_2_peak. Oxygen consumption, carbon dioxide production, and airflow were measured using MAX-II metabolic cart (AEI technologies, Landivisiau, France). Exhaled air was continuously sampled from a mixing chamber and averaged over 15 s-time windows. Oxygen consumption was measured in the first and last 15 min of the exercise test and used to confirm the relative oxygen consumption. If needed, small adjustments in workload were made to reach 70% of the V̇O_2_ peak of the individual. After the exercise protocol subjects went home, refrained from moderate to heavy physical activity, were instructed to keep low levels of light physical activity, and refrained from alcohol consumption until blood collection on the second study day. The habitual dietary intake of the study subjects was determined via a validated food frequency questionnaire (FFQ) that assessed dietary intake in the past four weeks [56]. The self-reported diets of the high-fit and low-fit subjects were similar with no significant differences in total daily energy intake, carbohydrate intake, protein intake, or fat intake (Appendix A).

### 4.4. Plasma and Serum Isolation

Plasma tubes were centrifuged for 10 min at 1200× *g* at 4 °C, and the supernatant (plasma) was collected, transferred to a new tube, and mixed. In the case of turbid plasma, samples were centrifuged again for 10 min at 1200× *g* at 4° to remove any insoluble matter. Plasma samples were snap-frozen in liquid nitrogen and afterwards cryopreserved at −80 °C. Serum tubes were centrifuged for 10 min at 1300× *g* at RT, and the supernatant (serum) was collected, transferred to a new tube, and mixed. In the case of turbid serum, samples were centrifuged again for 10 min at 1300× *g* at RT to remove any insoluble matter. Serum samples were snap-frozen in liquid nitrogen and afterwards cryopreserved at −80 °C. For biomarker and metabolic parameters analysis, plasma and serum samples were thawed on ice and individually mixed until a clear solution was reached. 

### 4.5. ELISAs in Serum and Plasma

Commercially available ELISA kits were used to analyse serum levels of the peptide hormones leptin, insulin, and adiponectin and the plasma levels of inflammatory and oxidative stress-related markers (tumour necrosis factor (TNF), IL6, IL10, CRP, the soluble monocyte differentiation antigen cluster of differentiation 14 (CD14), monocyte chemoattractant protein 1 (CCL2, better known as MCP1), soluble intercellular adhesion molecule 1 (ICAM1), lipopolysaccharide-binding protein (LBP), and oxidised low-density lipoprotein (oxidised LDL) according to the manufacturer’s instructions (Appendix A).

### 4.6. Proton NMR (^1^H NMR) in Plasma

EDTA-plasma samples were measured using the standardised, targeted high-throughput proton NMR (^1^H NMR) metabolomics from Nightingale Health (Nightingale Health Ltd., Helsinki, Finland, nightingalehealth.com/biomarkers). This platform provides simultaneous quantification of 162 individual metabolites and 87 metabolite ratios or sizes. For analysis of this study, all individual metabolites were selected, except for metabolite concentrations within lipoproteins or lipoprotein subclasses (e.g., ‘total lipids in VLDL’), and concentrations of clinical LDL cholesterol, remnant cholesterol, total cholesterol minus HDL cholesterol, and total branched-chain amino acids (BCAAs). All metabolite ratios or sizes were also excluded from the analysis. A complete list of the selected metabolites included in the analysis can be found in Appendix A. 

### 4.7. Proton NMR (^1^H NMR) in Serum

Serum samples were measured using targeted high throughput ^1^H NMR metabolomics at the EURECAT Technology Centre (Barcelona, Spain). For metabolite extraction, samples were placed in 2 mL 96 deep well plates using 200 mL methanol:water (8:1, for aqueous extraction), or 100 mL methyl-tert-butylether (MTBE):methanol:water (3:10:2, for lipidic extraction) in an automated fashion in the Bravo liquid handler (Agilent Technologies Santa Clara, CA, USA). Methanol and MTBE were purchased at Merck (Darmstadt, Germany). After extraction, solvents from the samples were removed using a speed vacuum concentrator and samples were stored at −80 °C until analysis. Some samples were lyophilised before ^1^H NMR analysis. For ^1^H NMR measurements, the hydrophilic extracts were reconstituted in 600 μL deuterium oxide (D_2_O, Deutero, Kastellaun, Germany) PBS (Sigma-Aldrich, St. Louis, MO, USA), 0.05 mM, pH 7.4, 99.5% D_2_O) containing 0.73 mM 3-(Trimethylsilyl) propionic-2,2,3,3-d4 acid sodium salt (TSP, Sigma-Aldrich), and the dried lipophilic extracts were reconstituted with a solution of deuterated chloroform (CDCL_3_)/deuterated methanol (CD_3_OD) (2:1, chloroform d-1 and methanol d-4 from Deutero) containing 1.18 mM tetramethylsilane (TMS, Sigma-Aldrich) and then vortexed. Both extracts were transferred into a 5 mm NMR glass tube for ^1^H NMR analysis. ^1^H NMR spectra were recorded at 300 K on an Avance III 600 spectrometer (Bruker, Billerica, Massachusetts, MA, USA) operating at a proton frequency of 600.20 MHz using a 5 mm PABBO gradient probe. Aqueous extracted samples were measured and recorded in processing number (procno) 11. For aqueous extracts one-dimensional ^1^H pulse experiments were carried out using the nuclear Overhauser effect spectroscopy (NOESY) presaturation sequence (RD–90°–t1–90°–tm–90° ACQ) to suppress the residual water peak, and the mixing time was set at 100 ms. Solvent presaturation with irradiation power of 160 mW was applied during the recycling delay (RD = 5 s) and mixing time. The 90° pulse length was calibrated for each sample and varied from 9.72 to 10.06 μs. The spectral width was 12 kHz (20 ppm), and a total of 256 transients were collected into 64 k data points for each ^1^H spectrum. Lipidic extracted samples were measured and recorded in procno 22. In the case of lipophilic extracts, a 90° pulse with presaturation sequence (zgpr) was used to suppress the water residual signal of methanol. An RD of 5.0 s, with an acquisition time of 2.94 s, was used. The 90° pulse length was calibrated for each sample and varied from 9.92 to 10.04 μs. After four dummy scans, a total of 128 scans were collected into 64 K data points with a spectral of 18.6 ppm. The exponential line broadening applied before the Fourier transformation was 0.3 Hz. The frequency domain spectra were phased, baseline-corrected, and referenced to TSP or TMS signal (d = 0 ppm) using TopSpin software (version 3.6, Bruker). All acquired ^1^H NMR were compared to standards of the pure selected compounds from the AMIX spectra database (Bruker), HMDB, and Chenomx databases for metabolite identification. In addition, we assigned metabolites by ^1^H–^1^H homonuclear correlation (COSY and TOCSY) and ^1^H–^13^C heteronuclear (HSQC) 2D NMR experiments and by correlation with pure compounds run in-house when needed. After pre-processing, specific ^1^H NMR regions identified in the spectra were integrated using the AMIX 3.9 software package. Curated identified regions across the spectra that were integrated using the same AMIX 3.9 software package were exported to Excel to evaluate the robustness of the different ^1^H NMR signals and to calculate the concentrations.

### 4.8. LC-MS/MS in Plasma

Plasma acylcarnitines were quantified or semi-quantitated in plasma by LC-MS/MS. Plasma samples were thawed at 4 °C and 30 μL samples were mixed with 270 μL 100% methanol containing the set of labelled internal standards (see Appendix A). The mixture was vortexed for 15 s and centrifuged for 10 min at 3800× *g* at 4 °C. The supernatant was transferred into a new plate and injected onto a Kinetex 2.6 μm Polar C18 column, 100 Å, 150 × 2.1 mm (Phenomenex, Torrance, CA, USA) using a UHPLC 1290 Infinity II Series system coupled to a QqQ/MS 6470 Series system (Agilent Technologies, Santa Clara, CA, USA). Metabolite extraction was carried out with a semi-automated process using Agilent Bravo Automated Liquid Handling Platform (Agilent Technologies, Santa Clara, CA, USA).

### 4.9. Statistical Analyses

Statistical analyses were performed using IBM SPSS Statistics for Windows (Version 25.0, IBM Corp, Armonk, NY, USA), and R (Version 4.1.2. R Core Team, Vienna, Austria). Graphs were created using GraphPad Prism (Version 8.0, Graphpad Software, CA, USA) and R. In total, 102 variables were included in the main analyses (RM-ANOVA, main effect analysis, PCA, heatmaps, correlation matrices). In the comparative analysis between identical metabolites in serum and plasma, 16 variables per platform (Nightingale or EURECAT) were included. 

#### 4.9.1. Data Representation and Transformation

Normality was checked using Shapiro-Wilk normality tests and tests for skewness and kurtosis. Normally distributed data are presented as mean ± SD and not normally distributed data are presented as median [interquartile range (IQR)]. For univariate analyses (repeated-measures analysis of variance (RM-ANOVA) and main effect analysis), not normally distributed data was transformed (log, inverse, square, inverse square root). For multivariate analyses (principal component analysis (PCA), hierarchical clustering and heatmap plotting, and correlation matrix analyses) all data was range scaled using the formula (x − min(x))/(max(x) − min(x)) [57] because all biomarkers and metabolic parameters were measured in different units. Scaling resulted in a value ranging from 0–1 for every variable but preserved the dynamic range within each biomarker and metabolic parameter. One sample on the EURECAT platform did not pass the quality assurance tests during ^1^H NMR analysis and was excluded from the analysis, resulting in N = 14 samples for the low-fit and N = 15 samples for the high-fit group for some analytes. 

#### 4.9.2. Bivariate Tests, RM-ANOVA, and Main Effects Analysis

Subject characteristics were compared using a Student’s unpaired *t*-test or Mann-Whitney U test. RM-ANOVA was used to study the effect of fitness level (between-subjects factor) and the effect of a recent bout of exercise (within-subjects factors) on single biomarker and metabolic parameter levels and the interaction between these two factors. All assumptions for RM-ANOVA were met. Partial eta square (η^2^) is given per effect as a measure of effect size. Since our study includes two repeated measures and the non-parametric alternative for a RM-ANOVA (Friedman ANOVA) requires three repeated measures, the six variables that did not achieve normality after data transformation were analysed using non-parametric bivariate analyses. Mann-Whitney U tests on the ranked baseline values were used to study the fitness level effect and on the ranked difference between baseline and post-exercise values to study the interaction effect. Wilcoxon-Signed rank tests on the ranked baseline and post-exercise values were used to study the exercise effect. No partial effect size measure could be calculated for these non-parametric tests. Raw *p*-values were corrected for multiple testing using Benjamini-Hochberg correction in the R package ‘FSA’ and a false discovery rate (FDR) was set at 10%. FDR-corrected *p*-values < 0.10 were considered statistically significant. None of the interactions between fitness level and the recent bout of exercise (P_group*exercise_) were <0.10 and the main effects of fitness level and the recent bout of exercise were therefore also analysed in a model without the interaction term (Appendix A).

#### 4.9.3. PCA, Hierarchical Clustering, and Heatmap Plotting

For PCA, the covariance matrix was computed, eigenvector decomposition was performed for principal component identification, and the first and second largest principal components were plotted in a projection matrix, using the R packages ‘ggplot2’‘tidyverse’, factoextra’ and ‘FactoMineR’. Hierarchical clustering was performed using Euclidean distance as the dissimilarity measure and complete linkage as the similarity measure between the clusters using the hclust function from R. Heatmaps were generated using the R package ‘ComplexHeatmap’. 

#### 4.9.4. Correlation Analyses

Levels of identical metabolites measured in serum (at EURECAT) and plasma (Nightingale) were compared using Spearman rank (for not normally distributed data) or Pearson (for normally distributed data) correlations on the raw data (16 variables per platform) to compare relative as well as absolute values. Spearman rho (ρ) or Pearson r (r) are given as effect size measures and *p*-values < 0.05 were considered statistically significant. The correlation matrix was generated by performing Spearman rank correlation analyses for all biomarker and metabolic parameter pairs. All scaled biomarker and metabolic parameter data (102 variables) of high-fit and low-fit subjects at baseline and at post-exercise (Figure 5), or at baseline only (Appendix A), were included. The correlation analysis used all scaled biomarker and metabolic parameter values without considering the fitness level or recent exercise effect. The correlation matrix was generated using the hclust function from R and the R packages ‘corrplot’ and ‘Hmisc’, returning a correlation plot based on hierarchically clustered biomarkers and metabolic parameters. Significant correlations (*p* < 0.05) are depicted by coloured wells and non-significant correlations (*p* > 0.05) are left blank.

## Figures and Tables

**Figure 1 ijms-24-04202-f001:**
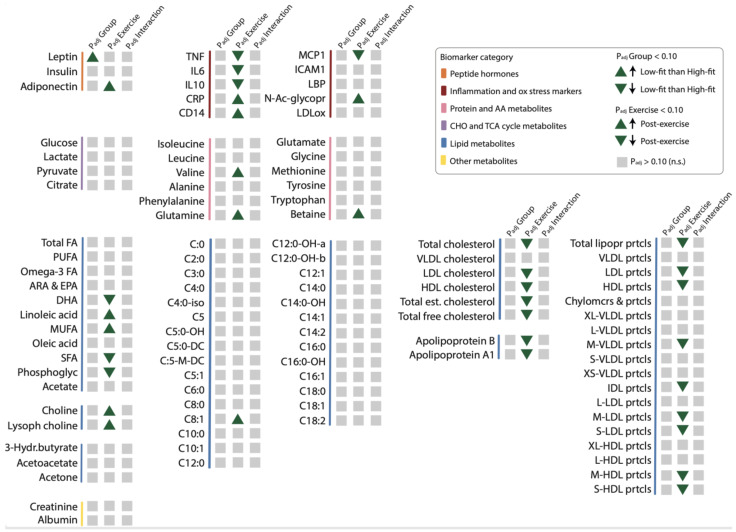
The effect of fitness level and a recent exercise bout on individual biomarker and metabolic parameter levels. Graphical summary representing the fitness level effect (P_adj_Group), recent exercise effect (P_adj_Exercise), and interaction effect (P_adj_Group*Exercise, shown as P_adj_Interaction) on individual biomarker and metabolic parameter levels within each functional biomarker and metabolic parameter category (indicated by colour). Significant fitness level effects (P_adj_group < 0.10) or recent exercise effects (P_adj_exercise < 0.10) are depicted by upward and downward green triangles that indicate the direction of the effect. Non-significant effects (P_adj_group, P_adj_exercise, or P_adj_group*exercise > 0.10) are depicted in grey squares (all interaction effects were not significant). The main effects (fitness level and recent exercise) and interaction effects were analysed using RM-ANOVA.

**Figure 2 ijms-24-04202-f002:**
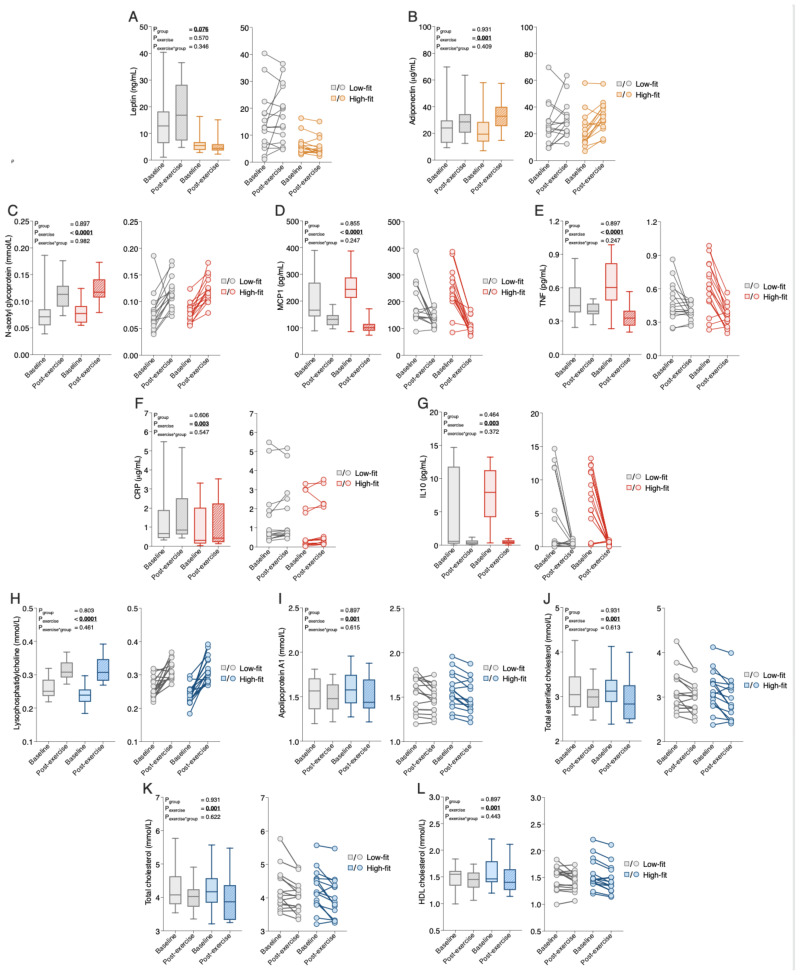
The response of the top five significantly regulated biomarkers and metabolic parameters within each biomarker and metabolic parameter category by fitness level, or a recent bout of exercise. (**A**,**B**) Median group levels (box plots, left) and individual levels (scatter plots, right) of leptin, (**A**) and adiponectin (**B**) in low-fit (N = 15, grey), and high-fit (N = 15, orange) females at baseline (transparent bars and dots) and after recent exercise (post-exercise; dashed bars and transparent dots). (**C**–**G**) Median group levels (box plots, left) and individual levels (scatter plots, right) of N-acetylglycoproteins (**C**), MCP1 (**D**), TNF I, CRP (**F**), and IL10 (**G**) in low-fit (N = 15, grey) and high-fit (N = 15, red) females at baseline (transparent bars and dots) and post-exercise (dashed bars and transparent dots). (**H**–**L**). Median group levels (box plots, left) and individual levels (scatter plots, right) of lysophosphatidylcholine (**H**), apolipoprotein A1 (**I**), total esterified cholesterol (**J**), total cholesterol (**K**), and HDL cholesterol (**L**) in low-fit (N = 15, grey (N = 14 for lysophosphatidylcholine)) and high-fit (N = 15, blue) females at baseline (transparent bars and dots) and post-exercise (dashed bars and transparent dots). The main effects (fitness level and recent exercise) and interaction effects were analysed using RM-ANOVA. Significant adj.*p*-values (<0.10) are indicated in underlined bold.

**Figure 3 ijms-24-04202-f003:**
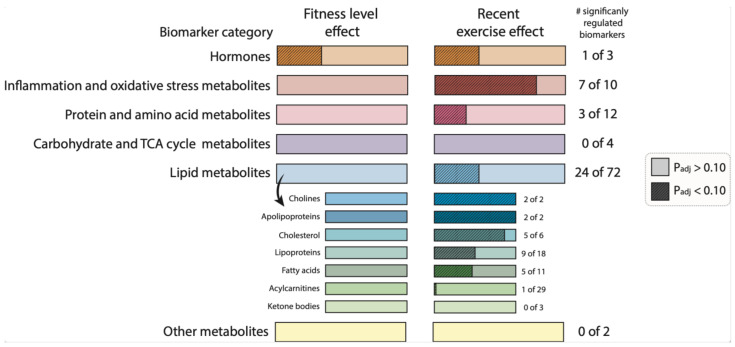
The effect of fitness level and a recent exercise bout on biomarkers and metabolic parameters category responses. Graphical summary representing the number of significantly regulated biomarkers and metabolic parameters between high-fit and low-fit females (fitness level effect, and left bars) and the number of significantly regulated biomarkers and metabolic parameters between baseline and post-exercise (recent exercise effect, right bars). Non-significant effects (adj.Pgroup or adj.Pexercise > 0.10) are depicted in light-coloured bars and significant effects (adj.Pgroup or adj.Pexercise < 0.10) are depicted in dark-coloured, dashed bars. The filled area is calculated relative to the number of biomarkers and metabolic parameters within the corresponding functional category.

**Figure 4 ijms-24-04202-f004:**
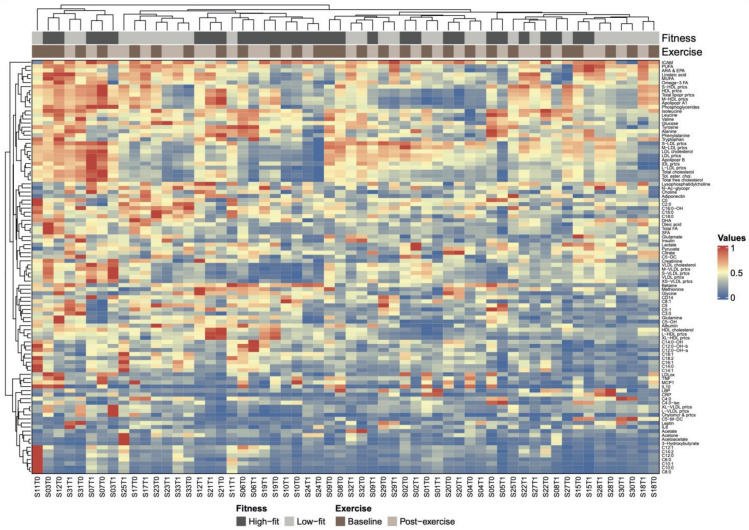
Heatmap of hierarchically clustered biomarkers and metabolic parameters and the association with fitness level and recent exercise. Heatmap based on hierarchical clustering of all 102 biomarkers and metabolic parameters based on Euclidean distance and complete linkage clustering. Biomarkers and metabolic parameters are clustered along the *y*-axis and individual subjects are clustered along the *x*-axis. Subject ID (S0x) and timepoint (T0 for baseline, T1 for post-exercise) are given for each subject. Subject IDs are coupled to fitness level (dark grey for high-fit and light grey for low-fit subjects) and timepoint (dark brown for baseline and beige for post-exercise). The colour scale represents low (dark orange) to medium (dark blue) biomarkers and metabolic parameter values.

**Figure 5 ijms-24-04202-f005:**
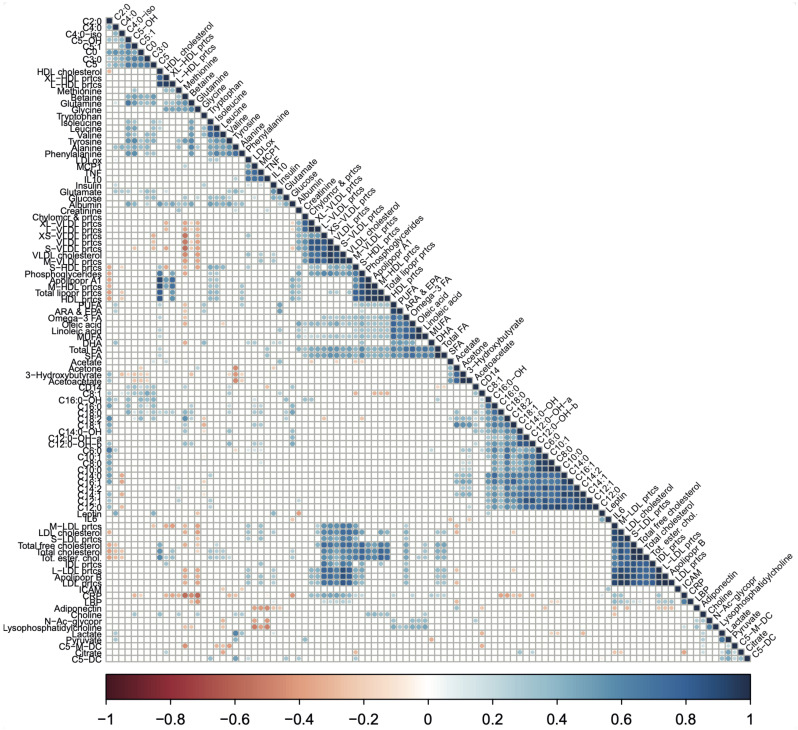
A correlation matrix showing the relationships between biomarker and metabolic parameter pairs. The correlation matrix based on Spearman correlation coefficients between biomarker and metabolic parameters pairs. Spearman rank correlation analysis was performed on the scaled biomarker and metabolic parameters values for all biomarker and metabolic parameters pairs using the combined data of high-fit and low-fit females at baseline and post-exercise. Relationships were considered statistically significant when *p* < 0.05. Significant relationships are indicated in red (negative correlation) or blue (positive correlation). Non-significant relationships (*p* > 0.05) are left blank.

**Table 1 ijms-24-04202-t001:** Subject characteristics.

	Low-Fit (N = 15)	High-Fit (N = 15)	*p*-Value
Age (years)	24.5 [22.9–25.6]	21.8 [21.6–23.7]	0.18
Ethnicity	Caucasian (10), Asian (1), Indo-pacific (4)	All Caucasian	
Weight (kg)	59.7 ± 7.1	61.2 ± 7.0	0.95
Height (m)	1.63 ± 0.08	1.68 ± 0.05	0.052
BMI (kg/m^2^)	22.4 ± 1.4	21.7 ± 1.9	0.28
Fat mass (% of weight)	28.7 ± 3.9	25.1 ± 4.4	0.025 *
Hemoglobin (mmol/L)	8.4 ± 0.6	8.5 ± 0.6	0.62
Use of birth control pill	6/15	7/15	
V̇O_2_peak (mL·kg ^−1^·min ^−1^)	35.0 [31.6–35.6]	50.4 [49.0–54.0]	<0.0001 ****
Baecke total score	7.3 ± 1.0	9.5 ± 0.8	<0.0001 ****
mV̇O_2_ recovery constant (% min ^−1^)	1.53 ± 0.46 ^#^	2.06 ± 0.57	0.018 *

BMI = body mass index, V̇O_2_peak = maximal oxygen consumption values, mV̇O_2_ = maximal oxygenation recovery constant in the *gastrocnemius* as a proxy for skeletal muscle mitochondrial capacity. ^#^ N = 11. Values are mean ± SD for normally distributed data, and median [IQR] for not normally distributed data. Significance was tested using unpaired two-tailed *t*-tests for all normally distributed data and using the Mann Whitney test for the not normally distributed VO2 peak data. * *p* < 0.05, **** *p* < 0.0001.

## Data Availability

An extensive, detailed table in spreadsheet format of the individual biomarkers is added as Appendix A. All underlying data are achieved on the servers of Wageningen University and are available upon reasonable request.

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
