# Peer review of "Single and Joined Behaviour of Circulating Biomarkers and Metabolic Parameters in High-Fit and Low-Fit Healthy Females"

_ijms, 2023, doi:10.3390/ijms24044202_

Round 1

Reviewer 1 Report

Janssen et al. aimed to explore differences in biomarker profile between high fit and low fit healthy female.

The idea is generally valid, yet this study is not nearly powered to demonstrate any difference, despite the fact that the authors used state of the art methodology, especially in choosing statistical tests.  The reason why most researchers usually do not explore differences in serum/plasma/urine biomarkers between subsets of healthy humans is simply because these differences are very subtle, and very large sample sizes are needed to demonstrate any difference reliably. In diseases, these differences are usually more pronounced and hence smaller sample is adequate. In this regard, I wonder if the authors calculated the sample size? I find it very misleading that the authors did not report small sample size in limitations section, given that the rest of their methodology is, as I already noted, state-of-the-art.

Perhaps if the subjects were identical twins, then this study would have much more sense. The fact that one group included 5 participants that are not Caucasian, unlike the other group which was all-Caucasian, further increases heterogeneity of the population and hence, biomarker variability.

It should be checked marker by marker, but I highly doubt that one day after exercise is reliable timeframe to establish “post-exercise” dynamic. I suppose that from 102 biomarkers some remain elevated the other day, but given the other causes of intra-individual variability these conclusions must not be taken for granted.

I wonder what is the rationale for including female participants only?  Since menstrual cycle may have affected a lot of biomarkers measured, I wonder if the authors taken that into consideration. 

Overall, the fact that study is not nearly adequately powered to demonstrate any conclusion prevents me from accepting this paper for further processing. 

Minor points

Table 1. P-values, alongside test used, should be presented for each variable in the table.

How were cut-offs for VO2peak established?

Reviewer 2 Report

The authors investigated how single biomarkers, functional biomarker categories and total biomarkers porfiles behave in young female adults of different aerobic fitness levels. Additionally, they investigate how these biomarkers are affected by recent exercise.

It is a well written and structured manuscript, currently of great interest.

The results are clear, well structured and detailed. In the discussion section also the strenghts and limitations of this trials were pointed out. Therefor there are only a few minor notes:

Introduction:

Line 61 to 80 - please delete information about study design (how many biomarkers were analysed; biomarker categories) from this introduction paragraph and add one summarizing sentence including information about the primary and secondary objective of the trial.

Supplementary Fig S2 

figure legend is not complete

Supplementary Fig S3

figure legend is not complete

Line 541 - please change 2.9 to 4.9 Statistical analyses

Materials and Methods

Line 399 please add a citation or explain a justification why you choose >47ml/kg/min for the high-fit group and <37ml/kg/min for the low-fit group.

Round 2

Reviewer 1 Report

No further comments.